# Relationship between Reactive Strength and Leg Stiffness at Submaximal Velocity: Effects of Age on Distance Runners

**DOI:** 10.3390/ijerph18136866

**Published:** 2021-06-26

**Authors:** Diego Jaén-Carrillo, Antonio Cartón-Llorente, Demetrio Lozano-Jarque, Alberto Rubio-Peirotén, Luis E. Roche-Seruendo, Felipe García-Pinillos

**Affiliations:** 1Department of Physiotherapy, Universidad San Jorge, Autovía Mudéjar, Km. 299, 50830 Villanueva de Gállego Zaragoza, Spain; djaen@usj.es (D.J.-C.); acarton@usj.es (A.C.-L.); dlozano@usj.es (D.L.-J.); leroche@usj.es (L.E.R.-S.); 2Department of Physical Education, Sports and Recreation, Universidad de La Frontera, Francisco Salazar, Temuco 1145, Chile; fegarpi@gmail.com; 3Department of Physical Education and Sport, University of Granada, 18011 Granada, Spain

**Keywords:** jumping, performance, running, stretch-shortening cycle, plyometric exercises

## Abstract

Background: Musculotendinous reactive strength is a key factor for the utilization of elastic energy in sporting activities such as running. AIM: To evaluate the relationship between musculotendinous reactive strength and lower-limb stiffness during running as well as to identify age-related differences in both variables. Methods: Fifty-nine amateur endurance runners performed three 20-cm drop jumps and a constant 3-min easy run on a motorized treadmill. Reactive strength index and dynamic lower-limb stiffness were calculated with a photoelectric cell system by jumping and running, respectively. Additionally, sit to stand difference in plantar arch height was assessed as a static lower-limb stiffness measure. The cluster analysis allows the comparison between younger and older runners. Results: No significant correlations were found between jumping reactive strength and running lower-limb stiffness. The younger group performed better at drop jumps (*p* = 0.023, ES = 0.82), whereas higher-but-no-significant results were found for reactive strength index and stiffness-related metrics. Conclusions: Musculotendinous vertical reactiveness may not be transferred to combined vertical and horizontal movements such as running.

## 1. Introduction

During running, the leg function resembles the behaviour of a spring which compresses and decompresses continually [1], being the body mass responsible for such leg-spring compression [2]. Mechanical energy is stored over the leg-spring compression, represented by the eccentric phase of stance, whereas the concentric phase of stance releases the stored energy as elastic energy [2]. Lower-limb stiffness (LLS) [3] and the stretch-shortening cycle (SSC) [4] are the two most important neuromuscular elements linked to elastic energy use.

Testing LLS in running gait includes the highest specificity for runners and may be calculated at the most suitable speed (i.e., race pace) for the athlete [5]. Both components of LLS, vertical stiffness (Kvert) and leg stiffness (Kleg), influence the regulation of spatiotemporal and kinematic variables in running. While Kvert describes the ratio of maximal force to the vertical displacement of the centre of mass (COM) when its lowest point is reached, Kleg describes the mechanical behaviour of the leg’s structural components (i.e., muscles, tendons, and ligaments) [6,7]. Increases in Kvert and Kleg are connected to both increased intensity and improved performance of a particular task [5]. The role played by the foot is essential in running since it is the ultimate part of the body in contact with the ground. It has been suggested that stronger foot muscles lead to a stiffer longitudinal arch [8] which is associated with increased running performance [9].

To date, little evidence regarding the effects of age on LLS has been reported [10,11]. In one study, a single-leg hopping task at different frequencies on a force plate combined with a motion capture system was proposed to establish differences regarding Kvert between children and young adults [10]. Absolute Kvert values were similar between adults and children but once normalised to body weight, children exhibited higher Kvert for all the frequencies proposed [10]. Additionally, these authors stated that children are able to actively stiffen their leg in one-leg hopping to counterbalance for their undeveloped stretch reflex, being able to hop faster [10]. Other study proposed a countermovement jump to establish Kleg differences between 10 elder and 10 younger physically active healthy males finding that the Kleg exhibited by the younger group was 29.3% higher [11]. To the best of our knowledge, no additional research has explored age differences on LLS based on its behaviour during running activities.

The capacity of a runner to employ efficiently the SSC and the capacity to store and release elastic energy by the musculotendinous units is represented as reactive strength [12]. The reactive strength index (RSI), defined as the ability to switch rapidly from an eccentric to a concentric contraction [12], can be measured from the ratio of flight time (FT) and ground contact time (CT) by drop jump (DJ) [13,14]. The correlations between the RSI and LLS using DJs have been investigated [15]. A positive correlation between RSI and Kvert was found, meaning that greater stiffness values were associated with greater RSI [15].

The contribution of RSI to LLS regulation in highly trained sprinters has been demonstrated [16]. In that study, the authors reported large associations between RSI, calculated through a 0.5 m DJ, and very large Kvert with maximum velocity [16]. However, to the authors’ knowledge, the connections between RSI and LLS at a slower velocity remain unknown as well as the effect of aging in both variables. Hence, this study is aimed at assessing the relationship between LLS during running and RSI while jumping in amateur endurance runners, as well as identifying possible age-related differences. We hypothesised that no correlations exist between LLS in running and RSI in jumping due to the principle of specificity of each task. Additionally, we hypothesised that the older group displays lower values for both LLS and RSI.

## 2. Materials and Methods

The current work is an observational cross-sectional study. Data were collected over only one session in the biomechanics laboratory of the University San Jorge during March and April 2019.

### 2.1. Participants

Fifty-nine injury-free amateur endurance runners voluntarily participated in this study (Table 1). All the participants were habitually shod runners and met the inclusion criteria: (i) Older than 18 years old, (ii) capable of running 10 km in less than 50 min (45.12 ± 4.42 min), (iii) at least two running sessions weekly. After receiving detailed information on the objectives and procedures of the study, each participant signed an informed consent form in order to participate, which complied with the ethical standards of the World Medical Association’s Declaration of Helsinki (2013). It was made clear that the participants were free to withdraw from the study at any time. Sport sciences students and amateur runners of local running clubs were recruited for the study and the sample was selected by convenience. The study was approved by the Institutional Review Board of University San Jorge, Zaragoza, Spain (Ref. Number 006-18/19).

### 2.2. Procedures

Participants completed a 3-min running trial on a motorised treadmill with a slope of 0% at 12 km·h^−1^ (HP cosmos Pulsar 4P; HP cosmos Sports & Medical, Gmbh, Nußdorf, Germany). The following procedure under the same conditions, and instructed by a researcher, was performed by every participant.

Before the start of the testing session, the participants completed a dynamic warm-up protocol that consisted of movement preparation (squatting, lunging, and hinging), 5-min stationary cycling, dynamic stretching, running drills consisting of skipping, counter-movement jump (CMJ), CMJ with bounce, and ankle jumps as this type of warm-up has been suggested to optimise jumping performance [17]. Each participant performed 3 maximal DJ at a height of 20 cm [14] and the best performance was considered for analysis. The landing zone was established between two transmitting-receiving bars belonging to the photoelectric cell system (OptoGait, Microgate, Bolzano, Italy) previously validated to measure the vertical jump height [18]. Measurements of FT (ms) and CT (ms) were recorded, and their ratio (RSI) [13] as RSI is found to be a valid and reliable indicator of explosive performance [19]. Participants had a 1-min rest between jumps [14]. To start, they were asked to ‘step out’ from the box, keeping their hands on their hips to minimise arm movement, and ‘to jump as high and as fast as possible’ on landing [20]. Each jump was analysed carefully and considered unacceptable in case that either the participants’ legs were not fully extended during the flight or they jumped forward off the landing zone.

Immediately after, an 8-min treadmill running accommodation program [21] was implemented by increasing speed by 1 km·h^−1^ every minute from 8 to 12 km·h^−1^. As spatiotemporal parameters reach a steady state soon [22], a 3-min running bout at a speed of 12 km·h^−1^ was proposed being 6–8 strides analysed [23]. Data were recorded for subsequent analysis.

### 2.3. Materials and Testing

As participants entered the laboratory, both body mass (kg) and height (cm) were determined using a weighing scale (Tanita BC-601; TANITA Corp., Maeno-Cho, Itabashi-ku, Tokyo, Japan) and a stadiometer (SECA 222; SECA Corp., Hamburg, Germany).

Total foot length (FL) and truncated foot length (TFL) were measured using the procedures outlined by Butler et al. [24]. First, linear dimensions of the unloaded right foot placed on an osteometric board were measured using sliding digital callipers with participants seated in a height-adjustable chair keeping their knees and hips under an alignment of 90° [25]. Feet, positioned 15 cm apart, were fixed in the heel cups. FL was measured from the most posterior part of the calcaneus to the most distal part of the longest toe. TFL is defined as the foot length from the most posterior part of the calcaneus to the centre of the medial joint space of the first metatarsal phalangeal joint [24]. The arch height index (AHI) was defined as the height of the arch at 50% total FL divided by TFL [26]. The same measures were done with participants standing in order to acquire loaded foot dimensions [25]. Moreover, the dorsal arch height difference between dorsal arch in a bipedal stance (AHIstand) and while sitting (AHIsit) was calculated, known as sit-to stand difference [26]. Similarly, arch stiffness was measured using the procedures described by Zifchock et al. [26].

Every measure was repeated 3 times and the average was computed and used for analysis. The static foot posture and foot mobility measures have reported moderate to good intrarater reliability (intraclass correlation coefficient = 0.81–0.99) and moderate to good interrater reliability (intraclass correlation coefficient = 0.58–0.99), respectively [25].

Measurements of the running spatiotemporal parameters of CT (the time the foot spends in contact with the ground) and FT (the time from the toes off to the initial contact of the same foot) [27] were acquired using the same photoelectric cell system (Microgate, Bolzano, Italy) described above, in which reliability was determined for the assessment of running spatiotemporal parameters and LLS [28]. The system was calibrated following the manufacturer’s recommendations and collected data at a sampling frequency of 1000 Hz placed on the edges of the treadmill surface for this study. The OptoGait system was linked to a laptop and the manufacturer software was used (version 1.12.1.0, Microgate, Bolzano, Italy). Filter parameters GAitR-In and GAitR-Out were set at 0_0 [29].

Running gait LLS reliability has been previously reported [30,31]. Both Kvert (kN/m), defined as the ratio of maximal force to the vertical COM displacement at the middle of the stance phase, and Kleg (kN/m), defined as the ratio of the maximal force in the spring to the maximum leg compression at the middle of the stance phase [6], were calculated using the Morin’s sine-wave method which has shown strong consistency with force plate methods [32].

### 2.4. Statistical Analysis

Descriptive data are presented as the mean and standard deviation (SD). The normality distribution of the data and homogeneity of variances were confirmed by Shapiro-Wilk’s and Levene’s tests, respectively (*p* > 0.05). In order to analyse the relationship between parameters, a partial correlation analysis, adjusted by age, was conducted. The following criteria were adopted to interpret the magnitude of correlations between measurement variables: <0.1 (trivial), 0.1–0.3 (small), 0.3–0.5 (moderate), 0.5–0.7 (large), 0.7–0.9 (very large), and 0.9–1.0 (almost perfect) [33]. Additionally, a cluster k-means analysis regarding the age of the subjects allows the authors to split up the whole group into two age-groups (younger group [YG] and older group [OG]). To explore the differences between age groups, an analysis of variance (ANOVA) was conducted for stiffness parameters (i.e., variables considering body mass), and an analysis of covariance (ANCOVA), considering the body mass as covariate, was performed for jumping performance parameters. The magnitude of the differences between values was also interpreted using the Cohen’s d effect size (ES) (between-group differences) [34]. Effect sizes are reported as: Trivial (<0.19), small (0.2–0.49), medium (0.5–0.79), and large (≥0.8) [34]. All statistical analyses were performed using the SPSS software version 25.0 (SPSS Inc., Chicago, IL, USA) and statistical significance was accepted at an alpha level of 0.05.

## 3. Results

The cluster analysis allows the authors to split up the whole group (n = 59) into two age-groups (younger group [YG, n = 19] and older group [OG, n = 40]). Significant between-group differences (*p* < 0.001) were found in age (YG: 23.06 ± 2.88 years old; OG: 40.76 ± 5.86 years old).

A comparative analysis between age groups for stiffness-related parameters is provided in Table 2. The ANOVA revealed no significant differences between YG and OG in static (i.e., arch stiffness, *p* = 0.084, ES = 0.39) and dynamic measures (i.e., Kvert and Kleg, *p* = 0.642 and 0.942, respectively, with trivial ES) of lower-limb stiffness. Nevertheless, the YG showed higher, though not significant, values in all parameters.

Table 3 includes a between-group comparison for DJ performance and RSI. The ANCOVA, adjusted by body mass, revealed significant differences between groups in DJ20 (*p* = 0.023, ES = 0.82), with better performance for the YG compared to OG. No significant differences were found in RSI (*p* = 0.154) although the ES was medium (ES = 0.63).

A partial correlation analysis, adjusted by age (Table 4), reported some significant relationships between lower-limb stiffness parameters. The arch stiffness significantly correlated with Kvert and Kleg (r = 0.272 and 0.264, *p* < 0.05, respectively), even though the magnitude of the correlations was small in both cases, and the Kvert reported significant, and almost perfect, correlation with Kleg (r = 0.944, *p* < 0.001). No significant relationships were found between RSI and lower-limb stiffness (r < 0.1, *p* ≥ 0.05).

## 4. Discussion

The present study sought to examine the relationship between lower-limb stiffness and reactive strength index in amateur endurance runners, as well as identify the possible age-related differences. The results described above allow us to state that no significant correlations exist between lower-limb stiffness and RSI. The results found for LLS, although non-significant, exhibit age-related differences showing a tendency towards higher values in the younger group. Similarly, the YG achieved significant higher values during DJ and no-significant greater RSI values, both adjusted by body mass. This suggests that although one may be reactive over the sagittal plane (i.e., DJ), it may not reflect performances requiring reactiveness in both sagittal and horizontal planes (i.e., running). This statement seems to be supported by the specificity principle that claims that the task’s demands define the type of SSC used and, hence, the RSI values [5,35,36], confirming consequently our hypothesis.

Therefore, as heightened arch stiffness has been associated with better distance running performance [9], the role of the LLS and SSC related to the use of elastic energy [3,4], as well as the relationship of both Kvert and Kleg with running performance [3,37] are well known. The mean values of arch stiffness (888.54 ± 507.30) here reported are supported by both Lieberman et al. (823.6 ± 223.7) [8] and Garcia-Pinillos et al. (947.7 ± 418.8) [9] in habitually shod runners. The similarity of the results between the present study and the two aforementioned studies relies on the same methods to evaluate arch stiffness and the very similar individual characteristics of the participants.

As already mentioned, the increased Kvert and Kleg were identified for YG, which is aligned with a previous work [11]. This study determined how Kleg adjusts according to age by comparing 10 young adults to old adults (24.3 ± 2 years and 68.6 ± 5 years, respectively) [11]. The authors stated that Kleg and its neuromuscular function of storing and realising elastic energy may decrease as we get older as significant differences in Kleg (*p* < 0.05) were found between groups (YG = 2.44 ± 0.52 kN/m; OG = 1.72 ± 0.78 kN/m) [11]. Our findings endorse the study by Liu et al. as our YG exhibited higher values for both Kvert (21.3 ± 7.09 kN/m) and Kleg (6.52 ± 3.15 kN/m) than our OG (20.37 ± 4.27 kN/m and 6.46 ± 1.9 kN/m, respectively). The differences in the magnitude of the values may be taken from two different perspectives. First, the difference of age between groups in the Liu et al. study is greater than ours, therefore, the difference between the values in both groups is greater. Then, Kleg values may be affected by the different sporting tasks proposed by the present study (i.e., running) and Liu et al. (i.e., CMJ) to assess LLS, supporting therefore our hypothesis that the principle of specificity of the task may determine the neuromuscular function. 

The use of DJ to calculate RSI has been previously reported [13,14]. The reactive strength may vary depending on several individual parameters such as anthropometry or chronological age [38]. Our findings endorse the suggestion by Suchomel et al. [38]. Greater RSI values have been shown by the younger group motivated by a significant (*p* < 0.05) better performance during DJ (27.18 ± 5.98 cm and 23.36 ± 4 cm, respectively). The results reported here slightly differ from those found by Beattie et al. [13]. While our participants, all amateur endurance runners, exhibited RSI values of 2.09 ± 0.5 (YG) and 1.81 ± 0.43 (OG), the Beattie et al. participants, the sample made of different sports shows 1.26 ± 0.24 (weaker athletes) and 1.50 ± 0.33 (stronger athletes). This discrepancy in RSI values might be attributed to, apart from the differences in the participants, the different drop jump heights (20 cm in the current study and 30 cm in the Beattie et al. study). Of note, however, our findings are supported by the Markwick et al. results [14]. Although in the present study amateur runners were assessed, very similar RSI values were obtained using a DJ in both studies, particularly regarding our younger group (age: 23.06 ± 2.88 years) and Markwick’s participants (age: 25.8 ± 3.5 years) [14]. Regarding these findings, it seems logical to affirm that the neuromuscular mechanism of reutilization of elastic energy, measured by RSI 20, decreased as a consequence of aging. Apparently, the assessment of RSI during DJ correlates highly to Kvert [15], meaning that RSI seems to reflect lower-limb stiffness in DJ, and, since stiffness-related age differences have been thoroughly reported, it might also explain the differences in RSI values between both age groups. 

The reactive strength and LLS have essential roles in sporting movements such as running or jumping [5,39]. The contribution of reactive strength to the regulation of LLS has been assessed at maximal velocity [16] showing no associations between LLS and RSI values. Both a stiffer leg at the touchdown and increased breaking forces in a DJ with Kvert when sprinting [16] have been related to the capability of using the lower-limb musculotendinous structures within the SSC, which seems to directly influence stiffness regulation at maximum velocity [16]. To the best of the authors’ knowledge, this relationship has not been assessed at submaximal velocities. Our age-adjusted partial correlation between RSI and LLS shows that arch stiffness significantly correlated with Kvert (r = 0.272, *p* < 0.05) and Kleg (r = 0.264, *p* < 0.05). Similarly, a significant and almost perfect correlation was found between Kvert and Kleg (r = 0.944, *p* < 0.001). However, no significant relationships were found between LLS and RSI (r < 0.12, *p* ≥ 0.05). Therefore, it can be argued that the underpinned principle under running and jumping tasks determines the neuromuscular behaviour of LLS and SSC in amateur distance runners allowing us to suggest that the one’s ability to exert reactive strength over a vertical movement (i.e., DJ) may not be transferred to combined vertical and horizontal movements (i.e., running).

Although both the SSC and the LLS play an outstanding role within the neuromuscular behaviour in the use of elastic energy in sporting tasks such as running and jumping, it has been shown how their primary function may be reduced due to aging affecting consequently performance.

Despite the insightful findings reported here, several methodological limitations need to be considered. The running protocol was executed on a motorised treadmill at 12 km·h^−1^. Then, participants used their own running shoes, ensuring therefore the acquisition of their regular performance. Although significant, the age of our participants may be considered at the same stage, thus, future research studies should contemplate groups separated by bigger age differences. Likewise, our sample was made exclusively of males, preventing us from establishing sex differences.

## 5. Conclusions

This study determined the age-related differences between RSI in a DJ and lower-limb stiffness during running, as well as their possible correlations. For the conditions here tested, no significant correlations were found between RSI and LLS showing that what can be reactive vertically (i.e., jumping) might not be horizontally (i.e., running) supported by the principle of specificity of each task. As hypothesised, the neuromuscular behaviour (i.e., SSC and LLS) might be jeopardized due to aging as our findings show that the older group exhibits lower values for RSI than the younger group. In order to minimise the aging effects, heavy resistance training and light-loaded explosive resistance training to increase agonist muscle activation and optimise the force development rate, respectively, can be proposed [40]. The addition of these recommendations twice per week to the runners’ training routine would help maintain the functionality of both SSC and LLS in distance runners [41,42].

## Figures and Tables

**Table 1 ijerph-18-06866-t001:** Participants descriptive data (mean ± SD).

Variable	All (*n* = 59)	Age		*p*-Value
		YG (*n* = 19)	OG (*n* = 40)	
Age (years old)	27.26 (8.47)	23.05 (2.88)	40.76 (5.86)	<0.001
Height (cm)	167.73 (7.25)	168.29 (7.47)	165.93 (6.37)	0.291
Body mass (kg)	61.25 (8.38)	62.38 (8.60)	57.57 (6.66)	0.060

YG: Younger group; OG: Older group.

**Table 2 ijerph-18-06866-t002:** Between-group (YG vs. OG) comparison in static (i.e., arch stiffness) and dynamic measures (i.e., vertical and leg stiffness) of lower-limb stiffness (±SD).

Variable	All (*n* = 59)	Age		*p*-Value (ES)
		YG (*n* = 19)	OG (*n* = 40)	
Arch stiffness	888.54 (507.30)	824.85 (368.17)	1093.29 (793.88)	0.084 (0.39)
Kvert (kN/m)	21.08 (6.50)	21.30 (7.09)	20.37 (4.27)	0.642 (0.18)
Kleg (kN/m)	6.51 (2.90)	6.52 (3.15)	6.46 (1.99)	0.942 (0.03)

YG: Younger group; OG: Older group; ES: Cohen´s d effect size; Kvert: Vertical stiffness; Kleg: Leg stiffness; SD: Standard deviation.

**Table 3 ijerph-18-06866-t003:** Drop jump (DJ) performance parameters (±SD) and reactive strength index regarding age.

Variable	All (*n* = 59)	Age ^		*p*-Value (ES)
		YG (*n* = 19)	OG (*n* = 40)	
DJ20 (cm)	26.27 (5.78)	27.18 (5.98)	23.36 (4.00)	0.023 (0.82)
RSI20	2.02 (0.50)	2.09 (0.50)	1.81 (0.43)	0.154 (0.63)

SD: Standard deviation; YG: Younger group; OG: Older group; ES: Cohen’s d effect size; DJ20: Jump height from a 20 cm drop jump; RSI20: Reactive strength index calculated from DJ20. ^ One-way analysis of covariance with body mass as covariates.

**Table 4 ijerph-18-06866-t004:** Partial correlation analysis (r coefficient), adjusted by age, between arch stiffness, lower-body stiffness during running, and reactive strength index obtained from drop jumping.

	Arch Stiffness	Kvert	Kleg	RSI20
Arch stiffness	1	0.272 *	0.264 *	−0.057
Kvert		1	0.944 **	0.076
Kleg			1	0.036
RSI20				1

* *p* < 0.05, ** *p* < 0.001. Kvert: Vertical stiffness; Kleg: Leg stiffness; RSI20: Reactive strength index calculated from a 20 cm drop jump.

## Data Availability

Data supporting reported results can be obtained on demand by contacting either the corresponding author or the first author of the current study.

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
