# Peer review of "Relationship between Reactive Strength and Leg Stiffness at Submaximal Velocity: Effects of Age on Distance Runners"

_ijerph, 2021, doi:10.3390/ijerph18136866_

Round 1
Reviewer 1 Report
General comments
The authors studied the association between lower-limb stiffness and reactive strength in students that are runners looking to possible age-related differences. The analysis performed is relevant and can be used in both clinical and sport contexts. However, I have several statistical concerns that are related to the way that the problem is stated. This could lead to different statistical analysis decisions. It seems that the authors were between the association of two parameters reactive force-related parameters (jump and running), and then decided to study for dependent variables if there were significant differences. This situation is explored during specific comments.
Specific comments
Abstract: When we read the title and statistical decisions, there is a kind of incongruence. Looking at the title, it seems that the analysis will be on the effect of age on the relationship between reactive strength and leg stiffness. This problem could ask for regression. However, there is a separate vision of this issue in the abstract (and discussion). First, “evaluate the relationship between musculotendinous reactive strength and lower-limb stiffness during running”, then “to identify age-related differences in both variables”. I think these two routes ran decisions that could be unappropriated. It seems that the rationale could be more explored. I will develop more in the following sections.
Introduction:
Page 2, line 45 & line 55: First, it is stated that there is “little evidence”, and after, “no additional research has explored”. I think that authors should rephrase in a simple sentence starting with “To the best of our knowledge”.
Page 2, line 61: It is the first time that “CT” appears; please write in full.
Page 2, lines 66 to 68: I don’t understand. There is an effect of age on these variables because it is unknown what happens at slower velocity? Rephrase, please.
Method:
Page 2, lines 86 and 87: I am totally for students of any age, without any prejudice. However, I was not expecting in a convenience sample of students to find a higher number (n = 40) of so many older students (40.76 ± 5.86). Were the participants all students from the university?
Page 2, line 88: it should be better “… approved by the Institutional Review Board of University San Jorge, Zaragoza, Spain (Ref. Number XXXXX).
Page 3, line 93: The authors wrote previously in the introduction about the “highest specificity for runners and may be calculated at the most suitable speed (i.e., race pace) for the athlete”. That is, studies investigating running and gait should adapt submaximal effort for each participant by using a self-paced rate. Instead, the authors have used the same speed for all subjects. Thus, the imposed speed corresponds to different intensities for each subject.
Page 3, line 117: Please replace “body weight” by “body mass” since the unit is in kilograms.
Page 4, line 155: Authors reported normal distribution; however, nothing is said regarding variances equality.
Page 4, line 166: Why the authors use Cohen’s d, instead of partial eta square (or even better the generalized eta squared), if an ANOVA was performed? It is not incorrect to use Cohen’s d in between design, but since the authors decided to include covariates and cluster analysis to define the groups, the use of explained variance family could be more appropriate. Also, if the age distribution was other than that of the present study, the number of subjects in each cluster could be different, and also the results.
Results:
Page 5, lines 195 to 196: Despite being mentioned in the statistical analysis, the authors should mention the different meanings of having an r = .264 and r = .944.
Discussion:
Page 5, lines 206 to 207: Please rephrase. It's forced. It seems that the authors want to have the expected results despite not having significant differences.
Page 6, line 243 to 248: the jump height is a very relevant aspect that can alter not just the ground reaction force but also the way neuromuscular activation takes place. Jump height, like speed, was not normalized in both studies. The implications of this should be mentioned.
Page 6, lines 248 to 256: Please, rephrase the sentence. Too long and confused.
Pages 6 and 7, lines 263 to 265: Please, rephrase the sentence. Too long and confused.
Page 7, lines 277 to 279: The limitation that should be stated regarding speed is the lack of normalization, such as self-paced running. Also, the division by clusters is dependent on sample distribution. That’s one of the reasons that I think the problem should be addressed just like the title does.
Page 7, line 268: Ok, but are these significant correlations strong?
Conclusion:
The authors continue as in the discussion. For example: “plyometric exercises which increase the ability of the musculotendinous units to store and release elastic energy [41] and enhance LLS might be beneficial “. The authors did not perform a randomized control trial to observe something like this. In the conclusion, I was expecting conclusions about the hypotheses that were proposed. Then, the impact of this study and practical applications. Much less to say that twice a week can be efficient. This kind of statement should be in the discussion section.
All the best
Author Response
We very much appreciate your constructive comments, useful information and your time. Thanks to this review, our manuscript was substantially improved. Responses to your comments are written in bold.
General comments
The authors studied the association between lower-limb stiffness and reactive strength in students that are runners looking to possible age-related differences. The analysis performed is relevant and can be used in both clinical and sport contexts. However, I have several statistical concerns that are related to the way that the problem is stated. This could lead to different statistical analysis decisions. It seems that the authors were between the association of two parameters reactive force-related parameters (jump and running), and then decided to study for dependent variables if there were significant differences. This situation is explored during specific comments.
Specific comments
Abstract
When we read the title and statistical decisions, there is a kind of incongruence. Looking at the title, it seems that the analysis will be on the effect of age on the relationship between reactive strength and leg stiffness. This problem could ask for regression. However, there is a separate vision of this issue in the abstract (and discussion). First, “evaluate the relationship between musculotendinous reactive strength and lower-limb stiffness during running”, then “to identify age-related differences in both variables”. I think these two routes ran decisions that could be unappropriated. It seems that the rationale could be more explored. I will develop more in the following sections.
Introduction
Page 2, line 45 & line 55: First, it is stated that there is “little evidence”, and after, “no additional research has explored”. I think that authors should rephrase in a simple sentence starting with “To the best of our knowledge”.
Done, thanks.
Page 2, line 61: It is the first time that “CT” appears; please write in full.
Done, thanks.
Page 2, lines 66 to 68: I don’t understand. There is an effect of age on these variables because it is unknown what happens at slower velocity? Rephrase, please.
It has been rephrased to improve its readability.
Method
Page 2, lines 86 and 87: I am totally for students of any age, without any prejudice. However, I was not expecting in a convenience sample of students to find a higher number (n = 40) of so many older students (40.76 ± 5.86). Were the participants all students from the university?
Sorry, there was missing information in that sentence. This has been completed properly. Thank you very much.
Page 2, line 88: it should be better “… approved by the Institutional Review Board of University San Jorge, Zaragoza, Spain (Ref. Number XXXXX).
It has been paraphrased as suggested. Thanks.
Page 3, line 93: The authors wrote previously in the introduction about the “highest specificity for runners and may be calculated at the most suitable speed (i.e., race pace) for the athlete”. That is, studies investigating running and gait should adapt submaximal effort for each participant by using a self-paced rate. Instead, the authors have used the same speed for all subjects. Thus, the imposed speed corresponds to different intensities for each subject.
As stated in the participants section (page 2, line 81), every runner should be able to run 10km in less than 50 minutes and that corresponds to 12 km·h-1. Therefore, we decided to run the protocol at that velocity.
Page 3, line 117: Please replace “body weight” by “body mass” since the unit is in kilograms.
Done, thanks.
Page 4, line 155: Authors reported normal distribution; however, nothing is said regarding variances equality.
Thanks to the reviewer for this point. The Levene´s test was conducted to confirm variances homogeneity. This information has been added to the text.
Page 4, line 166: Why the authors use Cohen’s d, instead of partial eta square (or even better the generalized eta squared), if an ANOVA was performed? It is not incorrect to use Cohen’s d in between design, but since the authors decided to include covariates and cluster analysis to define the groups, the use of explained variance family could be more appropriate. Also, if the age distribution was other than that of the present study, the number of subjects in each cluster could be different, and also the results.
The authors see the reviewer´s point but we do not agree. The type or model of effect size measure used is always a controversial topic and it can (though minimally most of cases) modify the data interpretation.
For this specific context, the authors consider the Cohen´s d is correct. Please, see the next manuscript:
* Olejnik, S., & Algina, J. (2000). Measures of effect size for comparative studies: Applications, interpretations, and limitations. Contemporary educational psychology, 25(3), 241-286.
Results
Page 5, lines 195 to 196: Despite being mentioned in the statistical analysis, the authors should mention the different meanings of having an r = .264 and r = .944.
The authors agree and the text has been accordingly modified.
Discussion
Page 5, lines 206 to 207: Please rephrase. It's forced. It seems that the authors want to have the expected results despite not having significant differences.
You are right. The sentence has been rephrased.
Page 6, line 243 to 248: the jump height is a very relevant aspect that can alter not just the ground reaction force but also the way neuromuscular activation takes place. Jump height, like speed, was not normalized in both studies. The implications of this should be mentioned.
We do not see your point. Jump height is not relevant to us. What really matters to our study is its relationship with stiffness
Page 6, lines 248 to 256: Please, rephrase the sentence. Too long and confused.
You are right. The sentence has been paraphrased to make it clearer.
Pages 6 and 7, lines 263 to 265: Please, rephrase the sentence. Too long and confused.
You are right. The sentence has been paraphrased to make it clearer.
Page 7, lines 277 to 279: The limitation that should be stated regarding speed is the lack of normalization, such as self-paced running. Also, the division by clusters is dependent on sample distribution. That’s one of the reasons that I think the problem should be addressed just like the title does.
Thank you for your comment but this is not relevant to our study since, as previously stated, what really matters here is the relationship with stiffness.
Conclusions
The authors continue as in the discussion. For example: “plyometric exercises which increase the ability of the musculotendinous units to store and release elastic energy [41] and enhance LLS might be beneficial “. The authors did not perform a randomized control trial to observe something like this. In the conclusion, I was expecting conclusions about the hypotheses that were proposed. Then, the impact of this study and practical applications. Much less to say that twice a week can be efficient. This kind of statement should be in the discussion section.
We totally agree. The entire paragraph has been re-written. Thanks for such insightful comment.
Reviewer 2 Report
Specify meaning of "CT" on line 61.
Explain the meaning of the sentence of line 66 “…calculated through a DJ of 0.5 m, and Kvert and very large with maximum speed”
In the cited articles (10 and 11) the age difference between the groups was much greater. In reference 10 the study was done between children and adults. In reference 11, the mean age of the elder group was 68.6, and that of the young group was 24.3. In your study, there is no great age difference between the groups, so perhaps that explains why there are no significant correlations between jumping reactive strength and running lower-limb stiffness.
Why were there twice as many volunteers in the sample of older people compared to younger ones?
It would have been interesting to also include women and homogenize the type of footwear, since its greater or lesser quality can influence the parameters measured.
Inclusion and exclusion criteria must be completed. Were all participants students or what percentage?
In table 1, include meaning of OG and YG.
Author Response
We very much appreciate your constructive comments, useful information and your time. Thanks to this review, our manuscript was substantially improved. Responses to your comments are written in bold.
Specify meaning of "CT" on line 61.
Done, thanks.
Explain the meaning of the sentence of line 66 “…calculated through a DJ of 0.5 m, and Kvert and very large with maximum speed”
Sorry, the sentence was incomplete. The missing information has been added.
In the cited articles (10 and 11) the age difference between the groups was much greater. In reference 10 the study was done between children and adults. In reference 11, the mean age of the elder group was 68.6, and that of the young group was 24.3. In your study, there is no great age difference between the groups, so perhaps that explains why there are no significant correlations between jumping reactive strength and running lower-limb stiffness.
We agree.
Why were there twice as many volunteers in the sample of older people compared to younger ones?
We were not able to recruit more participants for the younger group.
It would have been interesting to also include women and homogenize the type of footwear, since its greater or lesser quality can influence the parameters measured.
The authors agree but we were not able to recruit enough women to include them in the statistical analysis.
Inclusion and exclusion criteria must be completed. Were all participants students or what percentage?
This information has been added.
In table 1, include meaning of OG and YG.
Done, thanks.
Reviewer 3 Report
Here Jaén-Carrillo and colleagues conducted a survey on the relationship between reactive strength and leg stiffness at submaximal velocity.
The Reviewer identified multiple strengths in this paper including the design, the significance, the discussions, and the limitations.
However, there is one issue needs to be addressed in a revision.
Why the authors choose only one velocity at 12 km·h-1?
And did the authors tried other velocity points?
The authors should discuss and summarize this setup.
Author Response
We very much appreciate your constructive comments, useful information and your time. Thanks to this review, our manuscript was substantially improved. Responses to your comments are written in bold.
Comments to the Authors
Here Jaén-Carrillo and colleagues conducted a survey on the relationship between reactive strength and leg stiffness at submaximal velocity.
The Reviewer identified multiple strengths in this paper including the design, the significance, the discussions, and the limitations.
However, there is one issue needs to be addressed in a revision.
Why the authors choose only one velocity at 12 km·h-1?
And did the authors tried other velocity points?
The authors should discuss and summarize this setup.
Thank you very much for your kind words. As stated in the participants section (page 2, line 81), every runner should be able to run 10km in less than 50 minutes and that corresponds to 12 km·h-1. Therefore, we decided to run the protocol at that only velocity.